# Exploratory data on the clinical efficacy of monoclonal antibodies against SARS-CoV-2 Omicron variant of concern

**Fulvia Mazzaferri[1], Massimo Mirandola[1], Alessia Savoldi[1], Pasquale De Nardo[1], Matteo Morra[1], Maela Tebon[1], Maddalena Armellini[1], Giulia De Luca[1], Lucrezia Calandrino[2], Lolita Sasset[2], Denise D'Elia[3], Emanuela Sozio[3], Elisa Danese[4], Davide Gibellini[5], Isabella Monne[6], Giovanna Scroccaro[7], Nicola Magrini[8], Annamaria Cattelan[2], Carlo Tascini[3], MANTICO Working Group, Evelina Tacconelli[1]***

[1]Infectious Diseases Division, Department of Diagnostics and Public Health, University of Verona, Verona, Italy; [2]Infectious Disease Unit, Padova University Hospital, Padua, Italy; [3]Infectious Diseases Division, Department of Medicine, University of Udine and Azienda Sanitaria Universitaria Friuli Centrale, Udine, Italy; [4]Section of Clinical Biochemistry, Department of Neuroscience, Biomedicine and Movement, University of Verona, Verona, Italy; [5]Microbiology and Virology Unit, Department of Diagnostics and Public Health, University of Verona, Verona, Italy; [6]Viral genomics and transcriptomics Laboratory, Istituto Zooprofilattico Sperimentale delle Venezie, Legnaro, Italy; [7]Direzione Farmaceutico, Protesica, Dispositivi Medici, Regione del Veneto, Venice, Italy; [8]Italian Medicines Agency, Rome, Italy

**\*For correspondence:**
evelina.tacconelli@univr.it

**Group author details:**
MANTICO Working Group See page 13

## Abstract

**Background:** Recent in-vitro data have shown that the activity of monoclonal antibodies (mAbs) targeting severe acute respiratory syndrome coronavirus 2 (SARS-CoV-2) varies according to the variant of concern (VOC). No studies have compared the clinical efficacy of different mAbs against Omicron VOC.

**Methods:** The MANTICO trial is a non-inferiority randomised controlled trial comparing the clinical efficacy of early treatments with bamlanivimab/etesevimab, casirivimab/imdevimab, and sotrovimab in outpatients aged 50 or older with mild-to-moderate SARS-CoV-2 infection. As the patient enrolment was interrupted for possible futility after the onset of the Omicron wave, the analysis was performed according to the SARS-CoV-2 VOC. The primary outcome was coronavirus disease 2019 (COVID-19) progression (hospitalisation, need of supplemental oxygen therapy, or death through day 14). Secondary outcomes included the time to symptom resolution, assessed using the product-limit method. Kaplan-Meier estimator and Cox proportional hazard model were used to assess the association with predictors. Log rank test was used to compare survival functions.

**Results:** Overall, 319 patients were included. Among 141 patients infected with Delta, no COVID-19 progression was recorded, and the time to symptom resolution did not differ significantly between treatment groups (Log-rank Chi-square 0.22, p 0.90). Among 170 patients infected with Omicron (80.6% BA.1 and 19.4% BA.1.1), two COVID-19 progressions were recorded, both in the bamlanivimab/etesevimab group, and the median time to symptom resolution was 5 days shorter in the sotrovimab group compared with the bamlanivimab/etesevimab and casirivimab/imdevimab groups (HR 0.53 and HR 0.45, 95% CI 0.36–0.77 and 95% CI 0.30–0.67, p<0.01).

**Conclusions:** Our data suggest that, among adult outpatients with mild-to-moderate SARS-CoV-2 infection due to Omicron BA.1 and BA.1.1, early treatment with sotrovimab reduces the time to

recovery compared with casirivimab/imdevimab and bamlanivimab/etesevimab. In the same population, early treatment with casirivimab/imdevimab may maintain a role in preventing COVID-19 progression. The generalisability of trial results is substantially limited by the early discontinuation of the trial and firm conclusions cannot be drawn.

**Funding:** This trial was funded by the Italian Medicines Agency (Agenzia Italiana del Farmaco, AIFA). The VOC identification was funded by the ORCHESTRA (Connecting European Cohorts to Increase Common and Effective Response to SARS-CoV-2 Pandemic) project, which has received funding from the European Union's Horizon 2020 research and innovation programme under grant agreement number 101016167.

**Clinical trial number:** NCT05205759.

## Editor's evaluation

This paper will be of broad interest to clinicians and scientists in the area, providing clinical trial data on how the efficacy of monoclonal antibodies targeting SARS-CoV-2 varies according to the variant of concern. The clinical outcome data were consistent with previously reported in vitro data, which are being used to inform the clinical use of monoclonal antibodies.

## Introduction

Coronavirus disease 2019 (COVID-19), which is caused by severe acute respiratory syndrome coronavirus 2 (SARS-CoV-2), has spread globally and poses a major challenge to healthcare systems worldwide. A high incidence of hospitalisation and death due to COVID-19 has been reported among older patients and those with certain coexisting conditions, such as obesity, diabetes mellitus, cardiovascular disease, chronic obstructive pulmonary disease, and chronic kidney disease (*Petrilli et al., 2020*; *Huang et al., 2020*). The implementation of mass vaccination campaigns has markedly reduced the healthcare burden related to COVID-19. Nevertheless, SARS-CoV-2 vaccination rates differ considerably across countries, and growing evidence suggests a reduced efficacy of vaccines against new viral variants of concern (VOC) (*Cao et al., 2022*; *Planas et al., 2022*; *Dejnirattisai et al., 2022*; *Andrews et al., 2022*).

Therapeutic agents directed against SARS-CoV-2 have been developed to prevent the COVID-19 progression, especially addressing high-risk groups of patients. Neutralising monoclonal antibodies (mAbs) target the spike protein of SARS-CoV-2 that mediates viral entry into host cells (*Benton et al., 2020*). Based on the results of randomised placebo-controlled trials showing the efficacy in preventing COVID-19 progression, drug regulatory authorities, such as the US Food and Drug Administration (FDA), the European Medicines Agency, and the Italian Medicines Agency (AIFA), had granted the emergency use authorisation status for bamlanivimab 700 mg combined with etesevimab 1400 mg, casirivimab 600 mg combined with imdevimab 600 mg, and sotrovimab 500 mg to treat early COVID-19 in patients at high risk of progression (*Dougan et al., 2021*; *Weinreich et al., 2021*; *Gupta et al., 2021*).

To date, two randomised trials have compared the clinical outcomes of these mAbs in preventing severe COVID-19, showing similar effectiveness of bamlanivimab/etesevimab vs casirivimab/imdevimab in patients infected with the alpha VOC (*McCreary et al., 2022*) and casirivimab/imdevimab vs sotrovimab in patients infected with the Delta VOC, respectively (*Huang et al., 2022*).

This paper reports the results of the MANTICO trial, a non-inferiority randomised controlled trial comparing the clinical efficacy of routinely-used mAbs in a real-life setting of outpatients aged 50 or older with early mild-to-moderate COVID-19. The patient enrolment started in December 2021 and was interrupted after the publication of in-vitro evidence that two treatments under investigation (bamlanivimab/etesevimab and casirivimab/imdevimab) were not effective against the new emerging viral Omicron VOC (*Cao et al., 2022*; *Planas et al., 2022*; *Dejnirattisai et al., 2022*). The analysis is therefore restricted to 319 randomised patients, who were enrolled up to the interruption for possible futility, and was performed according to the SARS-CoV-2 VOC (Delta and Omicron).

## Methods

### Trial design

The trial was designed as a pragmatic, randomised, single-blind, non-inferiority, parallel group, multi-centre, and controlled trial. Eligible subjects were outpatients aged 50 years or older, presenting at three trial sites in Italy (Verona, Padua, and Udine) with a positive test (either direct antigen or nucleic acid SARS-CoV-2) and mild-to-moderate COVID-19 symptoms within 4 days of the onset (*COVID-19 Treatment Guidelines Panel, 2019*). COVID-19 symptoms included cough, nasal congestion, sore throat, feeling hot or feverish, myalgia, fatigue, headache, anosmia/ageusia, nausea, vomiting, and/ or diarrhoea (*U.S. Department of Health and Human Services Food and Drug Administration, 2022a*). Predefined exclusion criteria included a peripheral oxygen saturation level of 93% or less on room air, a respiratory rate of 30 or more breaths per minute, a heart rate of 125 or more beats per minute, and previous COVID-19 treatments with mAbs.

Sample-size estimation was based on the only available double-blind, randomised, placebo-controlled trial assessing the clinical efficacy of casirivimab/imdevimab (reference standard) in preventing COVID-19 progression in adult outpatients with early mild-to-moderate SARS-CoV-2 (*Weinreich et al., 2021*). This study showed that the hospitalisation related to COVID-19 or all-cause mortality occurred in 7 of 736 patients in the casirivimab/imdevimab 1200 mg group (1.0%) and in 24 of 748 patients in the placebo group (3.2%) (relative risk reduction, 70.4%; *Weinreich et al., 2021*). Therefore, 5% COVID-19 progression was assumed in the casirivimab/imdevimab group. 5% non-inferiority margin was considered clinically relevant by the expert opinion of infectious disease and clinical trial specialists involved in the protocol development, taking into account both the estimates of COVID-19 progression in the study population in the absence of early treatment with mAbs (20%; *Istituto Superiore di Sanità, 2021*) and the efficacy of the reference standard (*Weinreich et al., 2021*). Using these parameters, 420 patients per group were needed to achieve 90% power with a one-sided α level of.025, allowing for 5% dropout.

Participants were randomly assigned in a 1:1:1 ratio to receive a single intravenous infusion over a period of 1 hr, consisting of a combination of 700 mg of bamlanivimab and 1400 mg of etesevimab or 500 mg of sotrovimab or a combination of 600 mg of casirivimab and 600 mg of imdevimab. The study drugs were diluted to 250 mL with normal saline. Patients were masked to treatment group assignment. Randomisation was computer generated in permuted blocks with a stratification based on site. The allocated drug was revealed to the investigator using an online randomisation module within the REDCap data management system (*Harris et al., 2009*).

The trial was conducted in accordance with the principles of the Declaration of Helsinki, the international ethical guidelines of the Council for International Organisations of Medical Sciences, the International Council for Harmonisation Good Clinical Practice guidelines, and applicable laws and regulations. All patients or their legally authorised representatives provided written informed consent. This study is registered with ClinicalTrials.gov, NCT05205759.

### Outcomes

The composite primary outcome was the COVID-19 progression, defined as hospitalisation, need of supplemental oxygen therapy, or death from any cause through day 14. The presence of any of the three variables qualified the presence of the COVID-19 progression. Prespecified secondary outcomes were emergency department visits through day 28, all-cause mortality through day 28, duration of supplemental oxygen therapy, rate and duration of non-invasive ventilation and mechanical ventilation, and time to sustained patient-reported symptom resolution, which was defined as the absence of any symptom related to COVID-19 for at least 24 hr (*U.S. Department of Health and Human Services Food and Drug Administration, 2022b*).

### Predictors

The main predictor was the treatment regimen randomised at enrolment (bamlanivimab/etesevimab, casirivimab/imdevimab, and sotrovimab). All patients were assessed at baseline for the following predictors to be tested for association with the time to symptom resolution: age, sex, BMI, relevant comorbidities (diabetes for which medication was warranted, cardiovascular disease [hypertension, coronary artery disease, and congestive heart failure], chronic kidney disease, chronic liver disease, chronic pulmonary disease, active cancer, transplant, and other immunocompromising conditions),

SARS-CoV-2 serological status (anti-spike IgG), and SARS-CoV-2 vaccination status. The SARS-CoV-2 serological status was categorised as serum antibody-negative (if test results were negative), serum antibody-positive (if test results were positive), or other (inconclusive or unknown results). The SARS-CoV-2 vaccination status was categorised as not vaccinated, partial or complete primary COVID-19 vaccination series administered more than 120 days before the enrolment, complete primary COVID-19 vaccination series administered 120 days or less before the enrolment, and booster vaccination (*Andrews et al., 2022*). These categories were further collapsed as not vaccinated and partial or complete primary COVID-19 vaccination series administered more than 120 days before the enrolment vs complete primary COVID-19 vaccination series administered 120 days or less before the enrolment and booster vaccination.

## Procedures and tools

Outpatient visits were scheduled at baseline, 14±3 days and 30±3 days after the randomisation. Patients were considered lost to follow-up if they repeatedly did not participate in scheduled visits and could not be contacted by the investigators. Medical evaluation, vital signs recording, and laboratory tests were performed at each visit. If patients missed the visits, they were called by telephone to assess clinical conditions.

The SARS-CoV-2 serological status was assessed using LIAISON SARS-CoV-2 TrimericS IgG assay (DiaSorin), an indirect chemiluminescence immunoassay detecting IgG against the spike viral protein in its native trimeric conformation, which includes receptor-binding domain and N-terminal domain sites from the three subunit S1. According to the manufacturer's instructions, binding antibody units (BAU)/mL ≥33.8 were considered positive for anti-trimeric spike protein specific IgG antibodies.

Nasopharyngeal swabs were processed using MagMAX Viral/Pathogen Nucleic Acid Isolation Kit and KingFisher automated extraction system (ThermoFisher Scientific). Viral RNA was detected using COVIDSeq amplicon-based Next Generation Sequencing Test combined with COVIDSeq V4 Primer Pool (Illumina, Inc). Sequencing libraries were synthesised using automated Microlab STAR liquid handler (Hamilton Company). Pooled samples were quantified using Qubit 2.0 fluorometer (Invitrogen Inc). Next generation sequencing was performed in 150 PE mode on NextSeq 550 Sequencing System (Illumina, Inc) or MiSeq System (Illumina, Inc) using the NextSeq 500/550 Mid Output Kit v2.5 or the Miseq Reagent Kit v3, respectively.

## Statistical analysis

For continuous variables, mean and SD or median and IQR were calculated. For categorical variables, count and percentages were used. All outcome variables estimates were reported with 95% CI (95% CI). Wilcoxon–Mann–Whitney test was used to compare independent groups. The association between categorical variables was assessed using the Fisher's test. The product-limit method (Kaplan and Meier) was used to describe the time to symptom resolution. Kaplan-Meier estimator and Cox proportional hazard model were used to assess the bivariate association of independent variables with the time-dependent outcome. Kaplan-Meier curves were plotted to depict the association between each predictor and symptom persistence, and the Log-rank test was used to compare survival functions. Predictors associated with the time to symptom resolution with a probability <0.05 were considered significant. A two-sided test of less than 0.05 was considered statistically significant in all analyses. All statistical analyses were performed with the use of Stata Version 17.0 (College Station, TX: StataCorp LP).

## Results

The first patient was enrolled on 9 December 2021. Overall, 319 patients underwent randomisation by 20 January 2022 and were assigned to receive bamlanivimab/etesevimab (106 patients), sotrovimab (107 patients), or casirivimab/imdevimab (106 patients). No patients reported previous SARS-CoV-2 infections. No patients were lost to follow-up. VOC data were available for 311 patients: 170 (53.3%) were infected with Omicron and 141 (44.2%) with Delta. Eight (2.5%) patients were excluded from this analysis due to the lack of SARS-CoV-2 VOC identification. *Figure 1* shows the flow diagram of the progress through the trial phases. Baseline characteristics of the population by type of SARS-CoV-2 VOC are reported in *Table 1*.

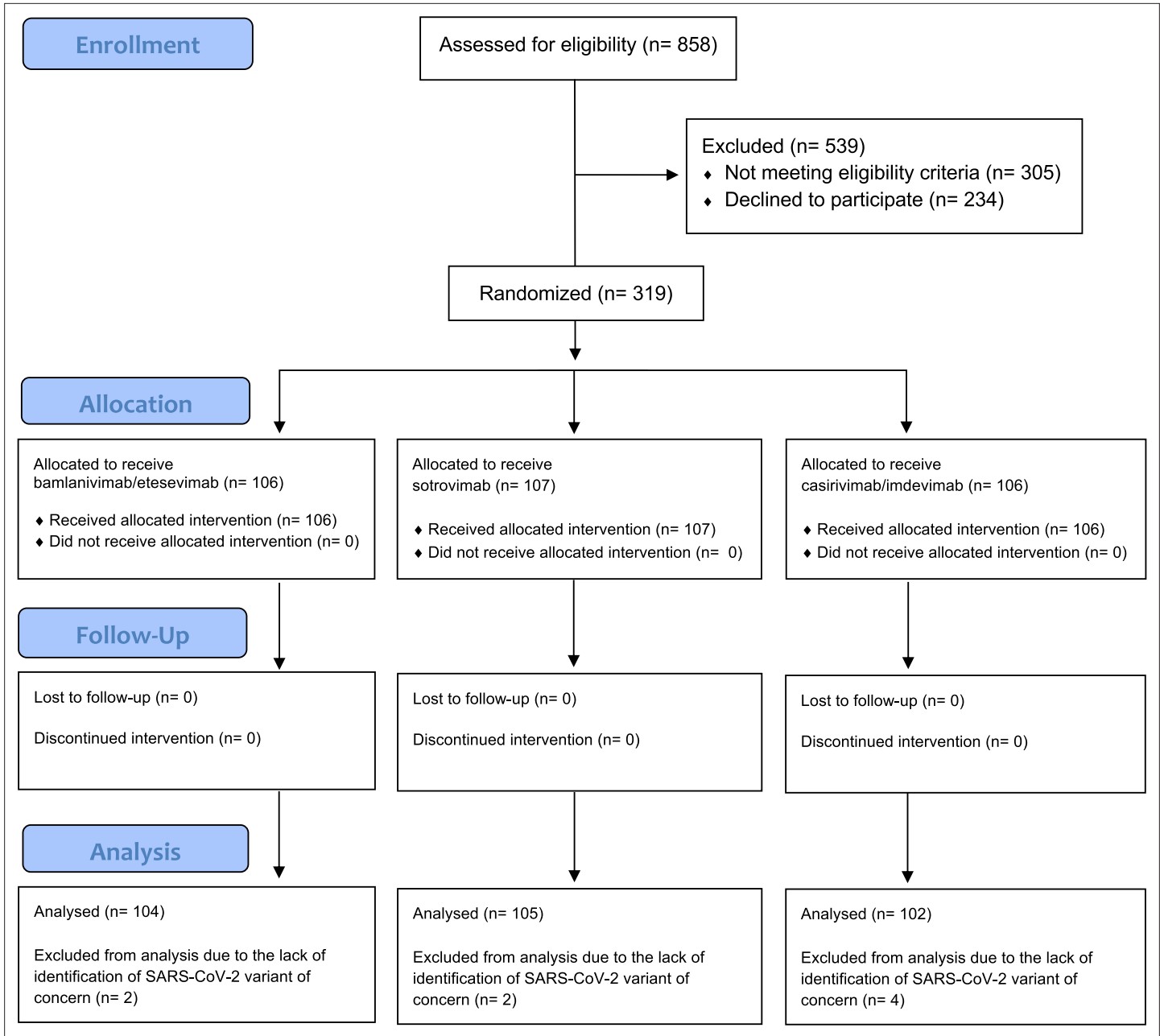

**Figure 1.** Flow diagram of the progress through the phases of the MANTICO trial according to the CONSORT statement.

Comparing symptoms at enrolment by VOC, anosmia/ageusia (p<0.001), nausea/vomiting (p<0.001), and feeling feverish or hot (p<0.01) were significantly more frequent among patients infected with Delta, while sore throat (p<0.001) was significantly more frequent among patients infected with Omicron. Serological positivity to anti-SARS-CoV-2 antibodies (p<0.001) and complete primary COVID-19 vaccination series within 120 days of the enrolment or booster vaccination (p<0.001) were significantly more frequent among patients infected with Omicron. *Table 2* shows the bivariate Cox regression of symptom resolution predictors by type of SARS-CoV-2 VOC. No predictors were associated with the time to symptom resolution in both SARS-CoV-2 VOC.

## Delta VOC

Baseline characteristics of 141 patients infected with Delta VOC by type of treatment are reported in *Table 3*. The main detected lineages were 34 AY.4 (24.1%), 33 AY.43 (23.4%), and 26 AY.122 (18.4%).

**Table 1.** Baseline characteristics of the overall study population by type of variant of concern.

| Characteristic | Delta N=141 | Omicron N=170 | p value |
|---|---|---|---|
| Sex (male) – n (%) | 69 (48.94) | 101 (59.41) | 0.068 |
| Age – median (IQR) (range) | 65.7 (15.4) (50–92) | 64.5 (14.8) (50–90) | 0.585 |
| Smoking status – n (%) | | | |
| Smoker | 8 (5.67) | 24 (14.12) | **0.015** |
| Former smoker | 32 (22.70) | 28 (16.47) | 0.194 |
| Non-smoker | 101 (71.63) | 118 (69.41) | 0.709 |
| BMI – n (%) | | | |
| ≤29 | 101 (71.63) | 132 (77.65) | 0.239 |
| ≥30 | 40 (28.37) | 38 (22.35) | 0.239 |
| SARS-CoV-2 serological status – n (%) | | | |
| Antibody-positive | 70 (49.65) | 134 (78.82) | **<0.001** |
| Antibody-negative | 68 (48.23) | 35 (20.59) | **<0.001** |
| Other | 3 (2.13) | 1 (0.59) | |
| Anti-SARS-CoV-2 vaccination status – n (%) | | | |
| 3 doses or 2 doses ≤120 days | 23 (16.31) | 66 (38.82) | **<0.001** |
| 1 or 2 doses ≥120 days or not vaccinated | 113 (80.14) | 99 (58.24) | **<0.001** |
| Other | 5 (3.55) | 5 (2.94) | |
| Comorbidities – n (%) | | | |
| Diabetes | 3 (2.13) | 6 (3.53) | 0.519 |
| Cardiovascular disease | 56 (39.72) | 61 (35.88) | 0.557 |
| Chronic kidney disease | 7 (4.96) | 9 (5.29) | 1.000 |
| Chronic liver disease | 3 (2.13) | 12 (7.06) | 0.061 |
| Chronic pulmonary disease | 16 (11.35) | 33 (19.41) | 0.061 |
| Immunocompromising conditions | 17 (12.06) | 35 (20.59) | **0.048** |
| Symptoms at enrolment – n (%) | | | |
| Cough | 96 (68.09) | 118 (69.41) | 0.807 |
| Nasal congestion | 69 (48.94) | 69 (40.59) | 0.169 |
| Sore throat | 32 (22.70) | 69 (40.59) | **0.001** |
| Feeling hot or feverish | 103 (73.05) | 99 (58.24) | **0.008** |
| Myalgia | 46 (32.62) | 54 (31.76) | 0.903 |
| Fatigue | 47 (33.33) | 75 (44.12) | 0.062 |
| Headache | 59 (41.84) | 60 (35.29) | 0.244 |
| Anosmia/ageusia | 39 (27.66) | 4 (2.35) | **<0.001** |
| Nausea/vomiting | 28 (19.86) | 11 (6.47) | **<0.001** |
| Diarrhoea | 15 (10.64) | 12 (7.06) | 0.314 |
| Serum C-reactive protein level – n | 136 | 161 | |
| Mean (SD), mg/L | 20.58 (29.00) | 14.29 (21.72) | **0.022** |

**Table 2.** Bivariate Cox regression of symptom resolution predictors by type of variant of concern.

| Predictor | Delta | | Omicron | |
| --- | --- | --- | --- | --- |
| | HR (95% CI) | p value | HR (95% CI) | p value |
| Gender | 0.80 (0.57–1.11) | 0.182 | 0.84 (0.61–1.14) | 0.257 |
| Age | 1.00 (0.98–1.02) | 0.952 | 1.00 (0.98–1.01) | 0.626 |
| BMI | 1.03 (0.72–1.50) | 0.855 | 1.17 (0.82–1.68) | 0.393 |
| SARS-CoV-2 serological status | 0.93 (0.67–1.31) | 0.690 | 0.82 (0.57–1.20) | 0.307 |
| Anti-SARS-CoV-2 vaccination status | 1.30 (0.83–2.04) | 0.257 | 0.91 (0.66–1.24) | 0.539 |
| Diabetes | 0.63 (0.34–1.18) | 0.150 | 1.19 (0.76–1.88) | 0.444 |
| Cardiovascular disease | 0.96 (0.69–1.35) | 0.831 | 0.85 (0.62–1.17) | 0.319 |
| Chronic kidney disease | 1.24 (0.58–2.66) | 0.581 | 1.12 (0.57–2.21) | 0.733 |
| Chronic liver disease | 2.42 (0.76–7.68) | 0.135 | 1.33 (0.74–2.40) | 0.341 |
| Chronic pulmonary disease | 0.78 (0.46–1.31) | 0.346 | 0.98 (0.67–1.43) | 0.902 |
| Immunocompromising conditions | 1.00 (0.60–1.66) | 0.989 | 0.80 (0.55–1.17) | 0.252 |

69 (48.9%) were male, median age was 65.7 years (IQR ±15.4), 115 (78.8%) had at least one comorbidity, 70 (49.6%) were serum antibody-positive at the enrolment, and 23 (16.3%) received complete primary COVID-19 vaccination series within 120 days of the enrolment or booster vaccination.

Primary and secondary outcomes of the study population infected with Delta VOC by type of treatment are reported in *Table 4* with the exclusion of time to sustained patient-reported symptom resolution. No COVID-19 progression was recorded in Delta infections. All-cause mortality through day 28 was the same as that through day 14. An emergency department visit without hospitalisation was observed once in one patient in the casirivimab/imdevimab group. This visit was deemed to be unrelated to COVID-19.

The median time to symptom resolution was 7 days (95% CI 6–13) in the bamlanivimab/etesevimab group, 10 days (95% CI 7–14) in the sotrovimab group, and 10 days (95% CI 7–15) in the casirivimab/imdevimab group, not differing significantly across the overall groups of treatment (Log-rank Chi-square 0.22, p 0.895) and for each comparison between treatment groups, namely bamlanivimab/etesevimab with casirivimab/imdevimab (Log-rank Chi-square 0.08, p 0.776), sotrovimab with casirivimab/imdevimab (Log-rank Chi-square 0.40, p 0.527), and bamlanivimab/etesevimab with sotrovimab (Log-rank Chi-square 0.01, p 0.92). *Figure 2A* shows the time to symptom resolution by type of treatment in the Delta study population. The Cox regression analysis confirmed the non-significantly different effects upon the time to symptom resolution between casirivimab/imdevimab (reference standard according to the original trial protocol) and both bamlanivimab/etesevimab and sotrovimab (HR 1.052 and HR 1.097, 95% CI 0.70–1.57 and 0.73–1.65, p 0.805 and 0.657, respectively).

## Omicron VOC
Baseline characteristics of 170 patients infected with Omicron VOC by type of treatment are reported in *Table 5*. The detected lineages were 137 (80.6%) BA.1 and 33 (19.4%) BA.1.1. 101 (59.4%) were male, median age was 64.5 years (IQR ±14.8), 135 (79.4%) had at least one comorbidity, 134 (78.8%)

**Table 3.** Baseline characteristics of the study population infected with Delta by type of treatment.

| Characteristic | Total N=141 | Sotrovimab N=43 | Bamlanivimab/ etesevimab N=48 | Casirivimab/ imdevimab N=50 |
|---|---|---|---|---|
| Sex (male) – n (%) | 69 (48.94) | 22 (51.16) | 21 (43.75) | 26 (52.00) |
| Age – median (IQR) (range) | 65.7 (15.4) (50–92) | 65.8 (16.4) (50–90) | 68.6 (11.8) (50–92) | 63.2 (12) (50–89) |
| Smoking status – n (%) | | | | |
| Smoker | 8 (5.67) | 2 (4.65) | 4 (8.33) | 2 (4.00) |
| Former smoker | 32 (22.70) | 8 (18.60) | 11 (22.92) | 13 (26.00) |
| Non-smoker | 101 (71.63) | 33 (76.74) | 33 (68.75) | 35 (70.00) |
| BMI – n (%) | | | | |
| ≤29 | 101 (71.63) | 29 (67.44) | 36 (75.00) | 36 (72.00) |
| ≥30 | 40 (28.37) | 14 (32.56) | 12 (25.00) | 14 (28.00) |
| SARS-CoV-2 serological status – n (%) | | | | |
| Antibody-positive | 70 (49.65) | 20 (46.51) | 29 (61.70) | 21 (43.75) |
| Antibody-negative | 68 (48.23) | 23 (53.49) | 18 (38.30) | 27 (56.25) |
| Other | 3 (2.13) | 0 | 1 (2.08) | 2 (4.00) |
| Anti-SARS-CoV-2 vaccination status – n (%) | | | | |
| 3 doses | 16 (11.35) | 6 (13.95) | 3 (6.25) | 7 (14.00) |
| 2 doses ≤120 days | 7 (4.96) | 2 (4.65) | 2 (4.17) | 3 (6.00) |
| 1 or 2 doses ≥120 days | 54 (38.30) | 14 (32.56) | 26 (54.17) | 14 (28.00) |
| Not vaccinated | 59 (41.84) | 19 (44.19) | 15 (31.25) | 25 (50.00) |
| Other | 5 (3.55) | 2 (4.65) | 2 (4.17) | 1 (2.00) |
| Comorbidities – n (%) | | | | |
| Diabetes | 3 (2.13) | 0 | 2 (4.17) | 1 (2.00) |
| Cardiovascular disease | 56 (39.72) | 18 (41.86) | 20 (41.67) | 18 (36.00) |
| Chronic kidney disease | 7 (4.96) | 1 (2.33) | 2 (4.17) | 4 (8.00) |
| Chronic liver disease | 3 (2.13) | 0 | 1 (2.08) | 2 (4.00) |
| Chronic pulmonary disease | 16 (11.35) | 6 (13.95) | 4 (8.33) | 6 (12.00) |
| Immunocompromising conditions | 17 (12.06) | 6 (13.95) | 6 (12.50) | 5 (10.00) |
| Symptoms at enrolment – n (%) | | | | |
| Cough | 96 (68.09) | 28 (65.12) | 36 (75.00) | 32 (64.00) |
| Nasal congestion | 69 (48.94) | 20 (46.51) | 22 (45.83) | 27 (54.00) |
| Sore throat | 32 (22.70) | 10 (23.26) | 8 (16.67) | 14 (28.00) |
| Feeling hot or feverish | 103 (73.05) | 31 (72.09) | 36 (75.00) | 36 (72.00) |
| Myalgia | 46 (32.62) | 11 (25.58) | 16 (33.33) | 19 (38.00) |
| Fatigue | 47 (33.33) | 13 (30.23) | 15 (31.25) | 19 (38.00) |
| Headache | 59 (41.84) | 15 (34.88) | 15 (31.25) | 29 (58.00) |
| Anosmia/ageusia | 39 (27.66) | 12 (27.91) | 15 (31.25) | 12 (24.00) |
| Nausea/vomiting | 28 (19.86) | 6 (13.95) | 9 (18.75) | 13 (26.00) |
| Diarrhoea | 15 (10.64) | 1 (2.33) | 5 (10.42) | 9 (18.00) |

*Table 3 continued on next page*

*Table 3 continued*

| Characteristic | Total N=141 | Sotrovimab N=43 | Bamlanivimab/ etesevimab N=48 | Casirivimab/ imdevimab N=50 |
|---|---|---|---|---|
| Serum C-reactive protein level – n | 136 | 41 | 46 | 49 |
| Mean (SD), mg/L | 20.58 (29.00) | 22.84 (33.70) | 25.27 (34.20) | 14.29 (15.99) |

were serum antibody-positive at the enrolment, and 66 (38.8%) received complete primary COVID-19 vaccination series within 120 days of the enrolment or booster vaccination.

Primary and secondary outcomes of the study population infected with Omicron VOC by type of treatment are reported in *Table 6* with the exclusion of time to sustained patient-reported symptom resolution. Two of 57 in the bamlanivimab/etesevimab group (3.5%) had COVID-19 progression leading to hospitalisation, and no COVID-19 progression was recorded in the casirivimab/imdevimab and sotrovimab groups. The primary reasons for the two hospitalisations were deemed to be related to COVID-19. Both patients admitted to hospital were serum antibody-negative at enrolment and underwent non-invasive mechanical ventilation at hospital admission. One of these patients, a man aged 71–75 who received three doses of SARS-CoV-2 vaccine and was affected by non-Hodgkin lymphoma under active chemotherapy and chronic heart failure, died 12 days after the symptom onset, 10 days after the administration of bamlanivimab/etesevimab, and 4 days after the hospital-isation. The other patient, a man aged 66–70 who was not vaccinated against SARS-CoV-2 and was affected by obesity (BMI, 31) and type 2 diabetes, was admitted 7 days after the symptom onset and 4 days after the administration of bamlanivimab/etesevimab; the length of his hospital stay was 22 days, including non-invasive mechanical ventilation for 13 days and low-flow oxygen therapy for 8 days. All-cause mortality through day 28 was the same as that through day 14.

An emergency department visit without hospitalisation was observed once in one patient in the bamlanivimab/etesevimab group. This visit was deemed to be unrelated to COVID-19.

The median time to symptom resolution was 12 days (95% CI 8–14) in the bamlanivimab/etesevimab group, 12 days in the casirivimab/imdevimab group (95% CI 9–16), and 7 days in the sotrovimab group (95% CI 6–9), differing significantly across the overall groups of treatment (Log-rank Chi-square 20.29, p 0.0001) and between sotrovimab and both bamlanivimab/etesevimab (Log-rank Chi-square 11.09, p 0.009) and casirivimab/imdevimab (Log-rank Chi-square 19.51, p 0.0001), whereas the compar-ison between bamlanivimab/etesevimab and casirivimab/imdevimab was not significant (Log-rank

**Table 4.** Efficacy outcomes of the study population infected with Delta by type of treatment with the exclusion of time to sustained patient-reported symptom resolution.

| Outcome | Total N=141 | Sotrovimab N=44 | Bamlanivimab/ etesevimab N=47 | Casirivimab/ imdevimab N=50 |
|---|---|---|---|---|
| Composite primary outcome – n (%) | 0 | 0 | 0 | 0 |
| Hospitalisation | 0 | 0 | 0 | 0 |
| Need of supplemental oxygen therapy | 0 | 0 | 0 | 0 |
| Death from any cause through day 14 | 0 | 0 | 0 | 0 |
| Secondary outcomes | | | | |
| Emergency department visits through day 28 – n (%) | 1 (0.71) | 0 | 0 | 1 (2) |
| All-cause mortality through day 28 – n (%) | 0 | 0 | 0 | 0 |
| Duration of supplemental oxygen therapy – days | 0 | 0 | 0 | 0 |
| Rate of non-invasive ventilation – n (%) | 0 | 0 | 0 | 0 |
| Duration of non-invasive ventilation – days | 0 | 0 | 0 | 0 |
| Rate of mechanical ventilation – n (%) | 0 | 0 | 0 | 0 |
| Duration of mechanical ventilation – days | 0 | 0 | 0 | 0 |

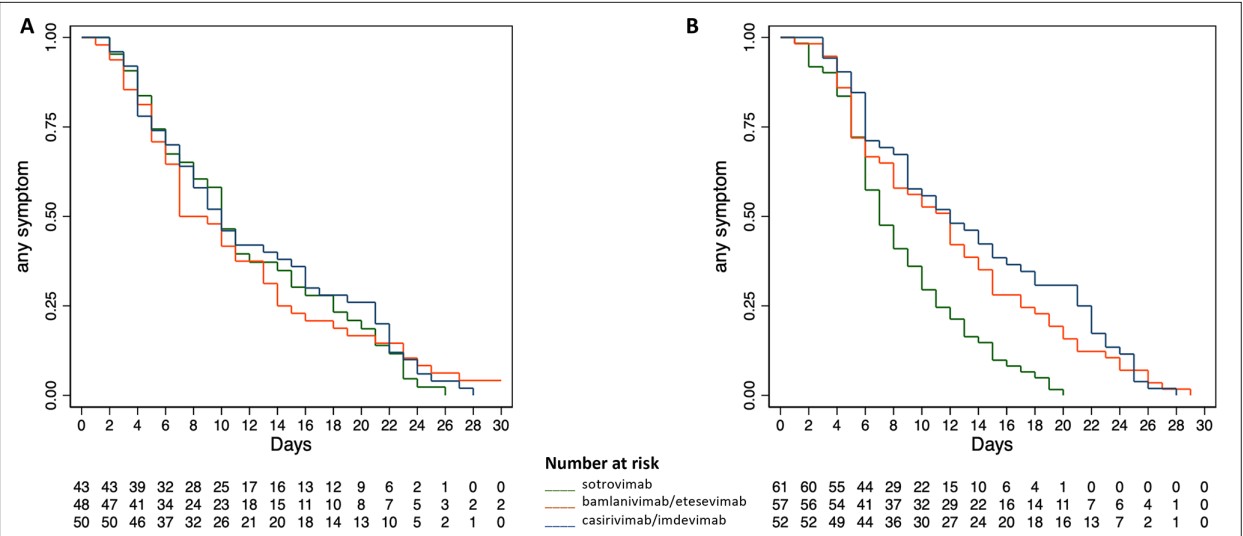

**Figure 2.** Time to symptom resolution by type of treatment in the study population infected with Delta (**A**) and Omicron (**B**).

Chi-square 0.63, p 0.427). The Cox regression analysis confirmed the significantly different effects upon the time to symptom resolution between sotrovimab and both bamlanivimab/etesevimab and casirivimab/imdevimab (HR 0.53 and HR 0.45, 95% CI 0.36–0.77 and 95% CI 0.30–0.67, p 0.001 and 0.0001, respectively). *Figure 2B* shows the time to symptom resolution by type of treatment in the Omicron study population. In each of the assessed subgroups (SARS-CoV-2 serological and vaccination status), sotrovimab showed a significantly shorter time to symptom resolution compared with bamlanivimab/etesevimab and casirivimab/imdevimab, as reported in *Table 7*.

## Discussion

During the SARS-CoV-2 pandemic, the paradigm of discovering and implementing mAb and antiviral treatments based on randomised controlled trials has lagged significantly behind the new evidence coming from in-vitro studies, which has driven clinical recommendations causing ethical dilemmas on the continuation of ongoing trials. At the time of approving the MANTICO trial protocol (November 2021), casirivimab/imdevimab, bamlanivimab/etesevimab, and, later, sotrovimab were the only therapies recommended by the COVID-19 treatment guidelines for outpatients with mild-to-moderate COVID-19 at high risk of progressing to severe COVID-19. Delta was the SARS-CoV-2 dominant VOC worldwide, and the selection of the study mAbs was based on their in-vitro activity against the circulating variants and on the existing evidence of their clinical efficacy. Since mid-December 2021, the Omicron VOC has been spreading worldwide, rapidly becoming the dominant VOC. Preliminary in-vitro studies on Omicron demonstrated numerous mutations in the gene encoding the spike protein, predicting a markedly reduced susceptibility to bamlanivimab/etesevimab and casirivimab/imdevimab (*Cao et al., 2022*; *Planas et al., 2022*; *Dejnirattisai et al., 2022*). According to these findings, FDA and AIFA have revised the emergency use authorisation for bamlanivimab/etesevimab and casirivimab/imdevimab, halting their use, in line with the National Institutes of Health COVID-19 Treatment Guidelines Panel, which advised against the use of these mAbs due to reduced activity against Omicron and because real-time testing to identify rare, non-Omicron variants is not readily available (*National Institutes of Health, 2022*). Therefore, the study enrolment in a real-life outpatient setting was prematurely discontinued for possible futility, after the inclusion of barely one fourth of the predefined sample size (1260 patients). Nevertheless, the recruitment timeframe provided a unique opportunity to collect data on the clinical efficacy of bamlanivimab/etesevimab, casirivimab/imdevimab, and sotrovimab in patients infected with Omicron.

Overall, the three treatment groups appeared to be balanced with respect to the predictors of outcomes in both Delta and Omicron population, as expected under the randomised allocation design. As reported by previous studies, patients infected with Omicron, compared with patients

**Table 5.** Baseline characteristics of the study population infected with Omicron by type of treatment.

| Characteristic | Total N=170 | Sotrovimab N=61 | Bamlanivimab/ etesevimab N=57 | Casirivimab/ imdevimab N=52 |
|---|---|---|---|---|
| Sex (male) – n (%) | 101 (59.41) | 36 (59.02) | 30 (52.63) | 35 (67.31) |
| Age – median (IQR) (range) | 64.5 (14.8) (50–90) | 64.2 (15) (50–90) | 64.8 (14.6) (50–86) | 65.3 (14.8) (50–86) |
| Smoking status – n (%) | | | | |
| Smoker | 24 (14.12) | 6 (9.84) | 11 (19.30) | 7 (13.46) |
| Former smoker | 28 (16.47) | 9 (14.75) | 11 (19.30) | 8 (15.38) |
| Non-smoker | 118 (69.41) | 46 (75.41) | 35 (61.40) | 37 (71.15) |
| BMI – n (%) | | | | |
| ≤29 | 132 (77.65) | 53 (86.89) | 42 (73.68) | 37 (71.15) |
| ≥30 | 38 (22.35) | 8 (13.11) | 15 (26.32) | 15 (28.85) |
| SARS-CoV-2 serological status – n (%) | | | | |
| Antibody-positive | 134 (78.82) | 45 (73.77) | 45 (78.95) | 44 (84.62) |
| Antibody-negative | 35 (20.59) | 16 (26.23) | 11 (19.30) | 8 (15.38) |
| Other | 1 (0.59) | 0 | 1 (1.75) | 0 |
| Anti-SARS-CoV-2 vaccination status – n (%) | | | | |
| 3 doses | 62 (36.47) | 24 (39.34) | 19 (33.33) | 19 (36.54) |
| 2 doses ≤120 days | 4 (2.35) | 2 (3.28) | 1 (1.75) | 1 (1.92) |
| 1 or 2 doses ≥120 days | 57 (33.53) | 16 (26.23) | 22 (38.60) | 19 (36.54) |
| Not vaccinated | 42 (24.71) | 18 (29.51) | 13 (22.81) | 11 (21.15) |
| Other | 5 (2.94) | 1 (1.64) | 2 (3.51) | 2 (3.85) |
| Comorbidities – n (%) | | | | |
| Diabetes | 6 (3.53) | 2 (3.28) | 2 (3.51) | 2 (3.85) |
| Cardiovascular disease | 61 (35.88) | 18 (29.51) | 17 (29.82) | 26 (50.00) |
| Chronic kidney disease | 9 (5.29) | 4 (6.56) | 2 (3.51) | 3 (5.77) |
| Chronic liver disease | 12 (7.06) | 4 (6.56) | 5 (8.77) | 3 (5.77) |
| Chronic pulmonary disease | 33 (19.41) | 11 (18.03) | 15 (26.32) | 7 (13.46) |
| Immunocompromising conditions | 35 (20.59) | 15 (24.59) | 11 (19.30) | 9 (17.31) |
| Symptoms at enrolment – n (%) | | | | |
| Cough | 118 (69.41) | 42 (68.85) | 37 (64.91) | 39 (75.00) |
| Nasal congestion | 69 (40.59) | 28 (45.90) | 25 (43.86) | 16 (30.77) |
| Sore throat | 69 (40.59) | 22 (36.07) | 27 (47.37) | 20 (38.46) |
| Feeling hot or feverish | 99 (58.28) | 37 (60.66) | 32 (56.14) | 30 (57.69) |
| Myalgia | 54 (31.76) | 20 (32.79) | 18 (31.58) | 16 (30.77) |
| Fatigue | 75 (44.12) | 31 (50.82) | 20 (35.09) | 24 (46.15) |
| Headache | 60 (35.29) | 23 (37.70) | 20 (35.09) | 17 (32.69) |
| Anosmia/ageusia | 4 (2.35) | 1 (1.64) | 2 (3.51) | 1 (1.92) |
| Nausea/vomiting | 11 (6.47) | 4 (6.56) | 5 (8.77) | 2 (3.85) |
| Diarrhoea | 12 (7.06) | 5 (8.20) | 4 (7.02) | 3 (5.77) |

*Table 5 continued on next page*

*Table 5 continued*

| Characteristic | Total N=170 | Sotrovimab N=61 | Bamlanivimab/ etesevimab N=57 | Casirivimab/ imdevimab N=52 |
|---|---|---|---|---|
| Serum C-reactive protein level – n | 161 | 57 | 56 | 48 |
| Mean (SD), mg/L | 14.29 (21.72) | 12.65 (15.97) | 17.19 (31.07) | 12.87 (12.55) |

infected with Delta, were more likely to present with symptoms limited to the upper respiratory tract and to have pre-existing immunity, considering that Omicron is better equipped than Delta to infect people with pre-existing immunity (*Nyberg et al., 2022*).

Considering the time to symptom resolution, no differences in the effect between treatment groups were found in Delta infections, whereas sotrovimab seems to show a benefit in patients infected with Omicron BA.1 and BA.1.1. This benefit was consistent across all Omicron subgroups, regardless of the SARS-CoV-2 serology and vaccination status, confirming the preliminary in-vitro evidence on the mAbs activity against Omicron BA.1 and BA.1.1 (*Cao et al., 2022*; *Planas et al., 2022*; *Dejnirattisai et al., 2022*).

The COVID-19 progression was recorded in two patients infected with Omicron, who were both randomised to receive bamlanivimab/etesevimab. On the one hand, these findings seem consistent with recent in-vitro data showing that all study treatments were active against Delta, and both casirivimab/imdevimab and sotrovimab retained a residual neutralising activity against Omicron BA.1/BA.1.1, whereas bamlanivimab/etesevimab did not neutralise Omicron (*Takashita et al., 2022b*; *Iketani et al., 2022*; *Takashita et al., 2022a*; *Arora et al., 2022*). Nevertheless, the above-mentioned results are severely limited by the early discontinuation of the trial, and firm conclusions on the primary outcome parameters cannot be drawn. Furthermore, the observed rate of COVID-19 progression (2/319, 0.6%) was lower than the one used to inform the sample size calculation (5% in the casirivimab/imdevimab arm, reference standard; NCT05205759). This overestimation of the primary outcome could be influenced by the lower intrinsic-severity of Omicron, the high vaccination rate in Italy, and the prioritisation of the booster vaccination for the elderly (*Bhattacharyya and Hanage, 2022*). Another limitation of this study is the lack of data on the clinical efficacy of the study mAbs, as well as other commercially available early COVID-19 treatments (mAbs, such as tixagevimab/cilgavimab

**Table 6.** Efficacy outcomes of the study population infected with Omicron by type of treatment with the exclusion of time to sustained patient-reported symptom resolution.

| Outcome | Total N=170 | Sotrovimab N=61 | Bamlanivimab/ etesevimab N=57 | Casirivimab/ imdevimab N=52 |
|---|---|---|---|---|
| Composite primary outcome – n (%) | 2 (1.18) | 0 | 2 (3.51) | 0 |
| Hospitalisation | 2 (1.18) | 0 | 2 (3.51) | 0 |
| Need of supplemental oxygen therapy | 2 (1.18) | 0 | 2 (3.51) | 0 |
| Death from any cause through day 14 | 1 (0.59) | 0 | 1 (1.75) | 0 |
| Secondary outcomes | | | | |
| Emergency department visits through day 28 – n (%) | 1 (0.59) | 0 | 1 (1.75) | 0 |
| All-cause mortality through day 28 – n (%) | 2 (1.18) | 0 | 2 (3.51) | 0 |
| Duration of supplemental oxygen therapy – days | 4 (patient 1) 22 (patient 2) | 0 | 4 (patient 1) 22 (patient 2) | 0 |
| Rate of non-invasive ventilation – n (%) | 2 (1.18) | 0 | 2 (3.51) | 0 |
| Duration of non-invasive ventilation – days | 4 (patient 1) 13 (patient 2) | 0 | 4 (patient 1) 13 (patient 2) | 0 |
| Rate of mechanical ventilation – n (%) | 0 | 0 | 0 | 0 |
| Duration of mechanical ventilation – days | 0 | 0 | 0 | 0 |

**Table 7.** Cox regression to assess the difference between treatment effects upon the time to symptom resolution in selected subgroups of interest in the study population infected with Omicron.

| Subgroup | Sotrovimab HR | Bamlanivimab/etesevimab HR (95% CI) | p value | Casirivimab/imdevimab HR (95% CI) | p value |
|---|---|---|---|---|---|
| **SARS-CoV-2 serological status** | | | | | |
| Antibody-negative | 1 | 0.34 (0.16–0.75) | **0.008** | 0.41 (0.18–0.97) | **0.043** |
| Antibody-positive | 1 | 0.40 (0.22–0.71) | **0.002** | 0.32 (0.18–0.57) | **<0.001** |
| **Anti-SARS-CoV-2 vaccination status** | | | | | |
| 1 or 2 doses ≥120 days or not vaccinated | 1 | 0.47 (0.30–0.77) | **0.003** | 0.49 (0.30–0.82) | **0.006** |
| 3 doses or 2 doses ≤120 days | 1 | 0.50 (0.28–0.89) | **0.019** | 0.35 (0.19–0.62) | **<0.001** |

and bebtelovimab, and antiviral drugs, such as remdesivir, nirmatrelvir/ritonavir, and molnupiravir), against the currently circulating VOC (BA.2 subvariants, BA.4, or BA.5; *CoVariants, 2022*). Following an adaptive design in a real-life setting, the MANTICO trial is actively recruiting to compare the clinical efficacy of commercially available early COVID-19 treatments against the currently circulating VOC (tixagevimab/cilgavimab, nirmatrelvir/ritonavir, and sotrovimab; NCT05321394).

Additional clinical studies with an adequate sample size are required to determine whether casirivimab/imdevimab and sotrovimab are indeed effective in preventing COVID-19 progression due to Omicron infection. Should the role of casirivimab/imdevimab in preventing severe COVID-19 due to Omicron infections be confirmed, this mAb could represent a readily available and well-tolerated treatment option in case of shortages of mAbs supplies and contraindication to other early COVID-19 treatments.

The MANTICO trial provides the first data on the clinical efficacy of bamlanivimab/etesevimab, casirivimab/imdevimab, and sotrovimab against Omicron VOC. There is an urgent need for adaptive clinical trials comparing anti-SARS-CoV-2 treatments by the currently circulating VOC to promptly inform recommendations for the management of early COVID-19.

# Additional information

### Group author details

**MANTICO Working Group**
**Francesca Simbeni**: Infectious Diseases Division, Department of Diagnostics and Public Health, University of Verona, Verona, Italy; **Alessandro Castelli**: Infectious Diseases Division, Department of Diagnostics and Public Health, University of Verona, Verona, Italy; **Ilaria Dalla Vecchia**: Infectious Diseases Division, Department of Diagnostics and Public Health, University of Verona, Verona, Italy; **Chiara Konishi de Toffoli**: Infectious Diseases Division, Department of Diagnostics and Public Health, University of Verona, Verona, Italy; **Gaia Maccarrone**: Infectious Diseases Division, Department of Diagnostics and Public Health, University of Verona, Verona, Italy; **Marco Meroi**: Infectious Diseases Division, Department of Diagnostics and Public Health, University of Verona, Verona, Italy; **Chiara Perlini**: Infectious Diseases Division, Department of Diagnostics and Public Health, University of Verona, Verona, Italy; **Matilde Rocchi**: Infectious Diseases Division, Department of Diagnostics and Public Health, University of Verona, Verona, Italy; **Giulia Rosini**: Infectious Diseases Division, Department of Diagnostics and Public Health, University of Verona, Verona, Italy; **Laura Rovigo**: Infectious Diseases Division, Department of Diagnostics and Public Health, University of Verona, Verona, Italy; **Lorenzo Tavernaro**: Infectious Diseases Division, Department of Diagnostics and Public Health, University of Verona, Verona, Italy; **Amina Zaffagnini**: Infectious Diseases Division, Department of Diagnostics and Public Health, University of Verona, Verona, Italy; **Ilaria Currò**: Infectious Diseases Division, Department of Diagnostics and Public Health, University of Verona, Verona, Italy; **Ruth Joanna Davis**: Infectious Diseases Division, Department of Diagnostics and Public Health, University

of Verona, Verona, Italy; **Elena Agostini**: Infectious Disease Unit, Padova University Hospital, Padua, Italy; **Carla Benfatto**: Infectious Disease Unit, Padova University Hospital, Padua, Italy; **Nicola Bonadiman**: Infectious Disease Unit, Padova University Hospital, Padua, Italy; **Marco Canova**: Infectious Disease Unit, Padova University Hospital, Padua, Italy; **Giuseppe Lombardo**: Infectious Disease Unit, Padova University Hospital, Padua, Italy; **Daniele Mengato**: Infectious Disease Unit, Padova University Hospital, Padua, Italy; **Danilo Puntrello**: Infectious Disease Unit, Padova University Hospital, Padua, Italy; **Francesca Prataviera**: Infectious Diseases Division, Department of Medicine, University of Udine and Azienda Sanitaria Universitaria Friuli Centrale, Udine, Italy; **Tosca Semenzin**: Infectious Diseases Division, Department of Medicine, University of Udine and Azienda Sanitaria Universitaria Friuli Centrale, Udine, Italy; **Dario Carloni**: Infectious Diseases Division, Department of Medicine, University of Udine and Azienda Sanitaria Universitaria Friuli Centrale, Udine, Italy; **Giuseppe Lippi**: Section of Clinical Biochemistry, Department of Neuroscience, Biomedicine and Movement, University of Verona, Verona, Italy; **Davide Negrini**: Section of Clinical Biochemistry, Department of Neuroscience, Biomedicine and Movement, University of Verona, Verona, Italy; **Martina Montagnana**: Section of Clinical Biochemistry, Department of Neuroscience, Biomedicine and Movement, University of Verona, Verona, Italy; **Riccardo Cecchetto**: Microbiology and Virology Unit, Department of Diagnostics and Public Health, University of Verona, Verona, Italy; **Alice Fusaro**: Viral genomics and transcriptomics Laboratory, Istituto Zooprofilattico Sperimentale delle Venezie, Legnaro, Italy; **Francesco Bonfante**: Viral genomics and transcriptomics Laboratory, Istituto Zooprofilattico Sperimentale delle Venezie, Legnaro, Italy; **Elisa Palumbo**: Viral genomics and transcriptomics Laboratory, Istituto Zooprofilattico Sperimentale delle Venezie, Legnaro, Italy; **Edoardo Giussani**: Viral genomics and transcriptomics Laboratory, Istituto Zooprofilattico Sperimentale delle Venezie, Legnaro, Italy

## Competing interests

Lolita Sasset: Lolita Sasset has served as a paid consultant to Abbvie, Janssen, MSD, Gilead Sciences, Janssen, MSD and ViiV Healthcare; she does not have any financial competing interests with this study. Annamaria Cattelan: Annamaria Cattelan has served as a paid consultant to Abbvie, Janssen, MSD, and received research fundings from Gilead Sciences, Janssen, MSD and ViiV Healthcare; she does not have any financial competing interests with this study. Carlo Tascini: Carlo Tascini has received grants from Correvio, Biotest, Biomerieux, Gilead, Angelini, MSD, Pfizer, Thermofisher, Zambon, Shionogi, Avir Pharma and Hikma outside the submitted work in the last two years. MANTICO Working Group: The other authors declare that no competing interests exist.

## Funding

| Funder | Grant reference number | Author |
| --- | --- | --- |
| Agenzia Italiana del Farmaco, Ministero della Salute | | Evelina Tacconelli |
| Horizon 2020 Framework Programme | 101016167 | Evelina Tacconelli |

The funders had no role in study design, data collection and interpretation, or the decision to submit the work for publication.

## Author contributions

Fulvia Mazzaferri, Conceptualization, Data curation, Software, Formal analysis, Supervision, Methodology, Writing - original draft, Writing – review and editing; Massimo Mirandola, Conceptualization, Data curation, Software, Formal analysis, Supervision, Validation, Visualization, Methodology, Writing - original draft, Writing – review and editing; Alessia Savoldi, Pasquale De Nardo, Resources, Data curation, Software, Formal analysis, Supervision, Validation, Investigation, Visualization, Methodology, Writing - original draft, Writing – review and editing; Matteo Morra, Data curation, Software, Supervision, Validation, Investigation, Project administration, Writing – review and editing; Maela Tebon, Data curation, Software, Supervision, Validation, Investigation, Project administration; Maddalena Armellini, Data curation, Validation, Investigation, Project administration; Giulia De Luca, Data curation, Validation, Investigation; Lucrezia Calandrino, Lolita Sasset, Data curation, Supervision, Investigation; Denise D'Elia, Resources, Data curation, Supervision, Investigation; Emanuela Sozio, Resources, Data

curation, Formal analysis, Supervision, Investigation, Writing - original draft; Elisa Danese, Resources, Validation, Investigation; Davide Gibellini, Resources, Supervision, Validation, Investigation; Isabella Monne, Resources, Supervision, Investigation, Writing - original draft; Giovanna Scroccaro, Conceptualization, Resources, Supervision, Project administration; Nicola Magrini, Carlo Tascini, Conceptualization, Supervision, Writing – review and editing; Annamaria Cattelan, Resources, Supervision, Writing – review and editing; MANTICO Working Group, Data curation, Investigation, Validation; Evelina Tacconelli, Conceptualization, Resources, Supervision, Funding acquisition, Visualization, Methodology, Project administration, Writing – review and editing

### Author ORCIDs
Massimo Mirandola ⬚ http://orcid.org/0000-0002-2342-5867
Evelina Tacconelli ⬚ http://orcid.org/0000-0003-2010-4977

### Ethics
Clinical trial registration NCT05205759.
All recruited subjects provided the informed consent to participate to the MANTICO trial. The IRB approval was provided by the Ethics Committee of the National Institute for Infectious Diseases "Lazzaro Spallanzani" (468_2021) and by the Scientific Technical Committee of the Italian Medicines Agency (28 OCT 2021).

### Decision letter and Author response
Decision letter https://doi.org/10.7554/eLife.79639.sa1
Author response https://doi.org/10.7554/eLife.79639.sa2

---

## Additional files

### Supplementary files
• MDAR checklist
• Reporting standard 1. CONSORT checklist.

### Data availability
The trial dataset has been uploaded to the Dryad repository (https://doi.org/10.5061/dryad.tdz08kq2w). As per predefined protocol, personally identifiable information (such as gender, date of birth, age, and weight) has been removed from the dataset to keep the records completely anonymous. In addition, the dataset record order was randomised so the resulting dataset is a file very similar in terms of length, fields and content to the original version, except for row order which is now completely random and the record id variable deleted.

The following dataset was generated:

| Author(s) | Year | Dataset title | Dataset URL | Database and Identifier |
| --- | --- | --- | --- | --- |
| Tacconelli E | 2022 | Adaptive, Randomized, Non-inferiority Trial to Evaluate the Efficacy of Monoclonal Antibodies in Outpatients With Mild or Moderate COVID-19 | https://dx.doi.org/10.5061/dryad.tdz08kq2w | Dryad Digital Repository, 10.5061/dryad.tdz08kq2w |

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
