## [Editor Report]

This paper will be of broad interest to clinicians and scientists in the area, providing clinical trial data on how the efficacy of monoclonal antibodies targeting SARS-CoV-2 varies according to the variant of concern. The clinical outcome data were consistent with previously reported in vitro data, which are being used to inform the clinical use of monoclonal antibodies.

---

## [Decision Letter]

**Decision letter after peer review:**

Thank you for submitting your article "Exploratory data on the clinical efficacy of monoclonal antibodies against SARS-CoV-2 Omicron Variant of Concern" for consideration by *eLife*. Your article has been reviewed by 3 peer reviewers, and the evaluation has been overseen by a Reviewing Editor and Miles Davenport as the Senior Editor. The following individual involved in the review of your submission has agreed to reveal their identity: David Huang (Reviewer #3).

Essential revisions:

1) As this is a report of a randomised trial, please report according to CONSORT guidelines/checklist and provide a CONSORT figure.

2) Clearer presentation of primary and secondary outcome data.

3) Revision of the discussion and conclusion to ensure that all statements regarding the efficacy of interventions and interpretations of findings are supported and consistent with the data presented (see reviewer comments below for examples).

4) Inclusion of discussion of limitations of the study, including potential sources of bias

*Reviewer #1 (Recommendations for the authors):*

The authors followed STROBE guidelines. However, reporting an RCT and not an observational study, they should have followed CONSORT guidelines.

Could the authors explain why the type of administered mAb was blinded to the patient?

*Reviewer #2 (Recommendations for the authors):*

Thank you for the opportunity to review this manuscript. I provide the below feedback in an effort to improve the manuscript and in addition to my evaluation summary and public review.

The authors could be more consistent with regard to the description of the primary outcome (e.g. "COVID-19 progression" on lines 38/39 of the abstract cf "disease progression" on line 44).

On lines 46/47, the authors could consider editing the statement "two disease progressions were recorded in the bamlanivimab/etesevimab group" to specify that there were no cases of disease progression in the other groups.

On lines 54/55, the evidence for the claim re: Casirivimab/imdevimab is insufficient (see statements above) but even as it is currently written it is not quite accurate. The manuscript states "Casirivimab/imdevimab seems to maintain a role in preventing severe COVID-19 in the Omicron population" whereas it could be better worded in terms of preventing progression of COVID-19 in adults with mild-moderate COVID-19 due to the omicron variant of SARS-CoV-2.

The ClinicalTrials.gov entry for your trial says 320 participants were recruited. Is there a reason that this manuscript says 319?

Is there a CONSORT diagram? Is there a CONSORT checklist that could be submitted with this manuscript? I see there is a STROBE checklist, and I guess in some ways this is an observational (δ vs omicron) study housed within an RCT. But I think it would be useful to understand (and report) more about the recruitment to the RCT, even as an appendix, and include a CONSORT diagram.

Were any participants lost to follow-up? Were any data points missing?

Figure 1 would usually show the primary outcome, rather than a secondary outcome. I can see why the authors would choose to show the secondary outcome (time to symptom resolution) in the figure, as this is a useful way of showing those data. But the discussion is focused on a signal in the primary outcome (which I do not think is justified) and I think if the authors wish to focus on that (hypothesis-generating only) finding of only 2 cases of COVID-19 progression, only occurring in the bamlanivimab/etesevimab group, the readers should see how these groups 'differ' visually too.

Could the word 'survival' lead to misinterpretation of the data in Figure 1? I would suggest removing 'survival' from the figure title (currently: "Figure 1 – Survival time to symptom resolution by type of treatment in the study population infected with Δ". Same for Figure 2).

Could Figure 1 and Figure 2 be combined into a multi-panel figure so the data are next to each other for comparison?

On lines 222/223, this statement is redundant when you have presented the days of symptoms in each group "turning out to be 5 days shorter in the sotrovimab group compared to both bamlanivimab/etesevimab and casirivimab/imdevimab groups"

I was confused about the presentation of the statistical analyses of the time to symptom resolution by treatment assignment in the two cohorts (δ and omicron). For the δ data, an analysis was presented assessing whether the duration of symptoms differed by treatment assignment (log-rank chi-square), and then each arm vs arm comparison was also presented as log-rank chi-square (lines 189-194). Cox regression was then used to confirm no difference with casirivimab/imdevimab as the reference. For omicron, where the data suggest there may be a difference, we do not present the chi-square analyses (lines 220-225). Is there a reason for this? The data should be presented the same way for both VOC.

Lines 224-226 too strongly imply a benefit here with casirivimab/imdevimab. And could be read as suggesting both casirivimab/imdevimab and sotrovimab retain comparable activity in vitro. Of course, it depends on the assay used, etc. And these in vitro assays have significant limitations, in terms of their utility to make treatment decisions (and decisions re: guideline recommendations!). But I think this section could be re-worded to improve accuracy.

When this is published, we will be on to the next variant (BA.2, BA.4, etc), in most places we already are. Therefore, consideration of how the authors frame the results in that context, ensuring we do not necessarily imply that all variants termed 'omicron' are equal with regards to mAb neutralisation, will be important (and challenging).

The authors could consider more of a discussion on the paucity of clinical data on these mAbs and these variants from well-designed trials, and that we do not know whether the in vitro findings have informed so many treatment/guideline decisions, do correlate with in vivo activity. Is this the first study to report clinical outcomes randomising patients across δ and omicron with these mAb agents? The authors could consider referring to this study, the other large comparative effectiveness study comparing monoclonal antibodies (particularly if published soon). https://www.medrxiv.org/content/10.1101/2021.09.03.21262551v1

*Reviewer #3 (Recommendations for the authors):*

My overall review is in the public review. What follows is details. Also, authors should check and review extensively for improvements to the use of English.

Abstract

– Notes AIFA and ORCHESTRA funding, but I believe the electronic form says no funding.

Introduction

– Other comparative effectiveness trials have been conducted.

https://www.medrxiv.org/content/10.1101/2021.12.23.21268244v1

https://www.sciencedirect.com/science/article/pii/S1551714422001483

Methods

– Please describe which centers participated and where they were located.

Results

– 319 pts enrolled in 5 weeks, over the winter holidays is impressive.

– Leveraging of the ~50/50 δ/omicron breakdown is very nice.

– For completeness, would note the #s of all the secondary outcomes, even if they are very small in # or even zero.

Discussion

– Would note that there was a signal in only 1 secondary outcome – could note it is perhaps due to the fact that very few patients had "progression" – but that nonetheless this signal is from 1 of multiple examined outcomes

– On page 10, line 255, "significant benefit" is somewhat problematic – statistically for the reason noted above, and 5d less symptoms is of debatable significance if ultimate outcome is the same. Would use more objective/neutral words.

– On page 11, line 265 – what are the specific findings that support that all study treatments were active against Δ (when there was no "control" arm) – the fact that no disease progression occurred – if so, pls say so?

Also, what are the specific findings that support that C-I and S both retain activity against Omicron when S was superior to both C-I and B-E for symptom resolution, and C-I and B-E had the same 12d median symptom duration? Ie. what is evidence that C-I retained activity v Omicron?

– Would note that currently the dominant Omicron variants are now BA.2 and beyond, and for this reason, FDA revoked all EUAs for mAbs, except for bebtelovimb.

Tables + Figures

– Please shorten everything to 1 or 2 decimal points.

– The K-M curves are hard to read – try dropping the 95% CIs.

[Editors' note: further revisions were suggested prior to acceptance, as described below.]

Thank you for resubmitting your work entitled "Exploratory data on the clinical efficacy of monoclonal antibodies against SARS-CoV-2 Omicron Variant of Concern" for further consideration by *eLife*. Your revised article has been evaluated by Miles Davenport (Senior Editor) and a Reviewing Editor.

The manuscript has been improved but there are some remaining issues that need to be addressed, as outlined below:

As recommended by the reviewers, please edit the abstract to ensure the conclusion reflects the study design and data. The abstract should state that the trial was ceased early and the conclusion amended as per reviewer 2 comments below.

For the study methods, the original sample size calculation should be included in the methods section (rather than referring readers to clinical trial register).

*Reviewer #2 (Recommendations for the authors):*

Thank you for submitting your revised manuscript and for addressing the suggestions raised in previous reviews. The revised manuscript is clearer, in terms of the study design and the important limitations, and the conclusions are now better supported by the data.

Given the uncertainty, the limitations, etc, the conclusion of the abstract is important. I suggest a minor edit of this sentence:

"Casirivimab/imdevimab seems to maintain a role in preventing COVID-19 progression in adult outpatients with early mild-to-moderate SARS-CoV-2 infection due to Omicron."

Which could be changed to:

"Casirivimab/imdevimab may maintain a role in preventing COVID-19 progression in adult outpatients with early mild-to-moderate SARS-CoV-2 infection due to Omicron."

This is particularly true when you are not suggesting it is used for treatment (which could be implied by "seems to maintain a role") and your overarching conclusion is that adaptive clinical studies are required (although casirivimab/imdevimab is not included in the next phase of MANTICO, presumably because these data are only available now and the VOCs have continued to change).

*Reviewer #3 (Recommendations for the authors):*

Revisions are appropriate, however, manuscript revisions (esp Discussion revisions) have not extended to the Abstract.

For Abstract, I recommend:

1. Cut decimal places to two.

2. Also add to the 2nd Conclusion sentence the specific Omicron variants analyzed in this study. Or reword so it's clear that the specific variants apply to all Conclusion sentences / mAbs.

3. Add to Conclusion a sentence reflective of the "results are severely limited" sentence added to Discussion.

E.g., could note that time to symptom resolution was 1 of X secondary outcomes, and no conclusions could be made about the primary outcome.

Specific text up to the authors – my main residual concern is simply that the Abstract's Conclusion is considerably "stronger" than the manuscript's conclusion/discussion, and should be appropriately caveated.

---

## [Author Response]

Essential revisions:1) As this is a report of a randomised trial, please report according to CONSORT guidelines/checklist and provide a CONSORT figure.

The CONSORT flow diagram has been added as Figure 1 and the CONSORT checklist has been included among the uploaded files.

2) Clearer presentation of primary and secondary outcome data.

Primary and secondary outcome data have been reported in Table 4 (Δ variant) and Table 6 (Omicron variant) of the revised version of the manuscript (pages 20 and 22).

The previous Figure 1 and 2 have been combined into a multi-panel figure, now renamed Figure 2A/2B (without 95% CIs, as suggested by Reviewer 3).

3) Revision of the discussion and conclusion to ensure that all statements regarding the efficacy of interventions and interpretations of findings are supported and consistent with the data presented (see reviewer comments below for examples).

The discussion and conclusion have been revised accordingly (page 12).

4) Inclusion of discussion of limitations of the study, including potential sources of bias

Limitations of the study, including potential sources of bias, have been included in the discussion of the revised version of the manuscript (page 12, lines 290-301).

Reviewer #1 (Recommendations for the authors):The authors followed STROBE guidelines. However, reporting an RCT and not an observational study, they should have followed CONSORT guidelines.Could the authors explain why the type of administered mAb was blinded to the patient?

The CONSORT flow diagram has been added as Figure 1 and the CONSORT checklist has been included among the uploaded files.

The type of administered mAb was blinded to the patient to reduce the risk of performance bias.

Reviewer #2 (Recommendations for the authors):Thank you for the opportunity to review this manuscript. I provide the below feedback in an effort to improve the manuscript and in addition to my evaluation summary and public review.The authors could be more consistent with regard to the description of the primary outcome (e.g. "COVID-19 progression" on lines 38/39 of the abstract cf "disease progression" on line 44).

The primary outcome has been consistently reported as “COVID-19 progression” all throughout the revised version of the manuscript.

On lines 46/47, the authors could consider editing the statement "two disease progressions were recorded in the bamlanivimab/etesevimab group" to specify that there were no cases of disease progression in the other groups.

The sentence has been amended accordingly in the revised version of the manuscript (page 2, line 47).

On lines 54/55, the evidence for the claim re: Casirivimab/imdevimab is insufficient (see statements above) but even as it is currently written it is not quite accurate. The manuscript states "Casirivimab/imdevimab seems to maintain a role in preventing severe COVID-19 in the Omicron population" whereas it could be better worded in terms of preventing progression of COVID-19 in adults with mild-moderate COVID-19 due to the omicron variant of SARS-CoV-2.

The sentence has been amended accordingly in the revised version of the manuscript (page 2, line 55-56).

The ClinicalTrials.gov entry for your trial says 320 participants were recruited. Is there a reason that this manuscript says 319?

Thanks for pointing out this typo. The right number is 319, as reported in the manuscript. The ClinicalTrials.gov entry has been corrected.

Is there a CONSORT diagram? Is there a CONSORT checklist that could be submitted with this manuscript? I see there is a STROBE checklist, and I guess in some ways this is an observational (δ vs omicron) study housed within an RCT. But I think it would be useful to understand (and report) more about the recruitment to the RCT, even as an appendix, and include a CONSORT diagram.

The CONSORT flow diagram has been added as Figure 1 and the CONSORT checklist has been included among the uploaded files.

Were any participants lost to follow-up? Were any data points missing?

No patients were lost to follow-up and no data points were missing.

Figure 1 would usually show the primary outcome, rather than a secondary outcome. I can see why the authors would choose to show the secondary outcome (time to symptom resolution) in the figure, as this is a useful way of showing those data. But the discussion is focused on a signal in the primary outcome (which I do not think is justified) and I think if the authors wish to focus on that (hypothesis-generating only) finding of only 2 cases of COVID-19 progression, only occurring in the bamlanivimab/etesevimab group, the readers should see how these groups 'differ' visually too.

We thank the reviewer for the suggestion. Primary and secondary outcomes have been reported in Table 4 (Δ variant) and Table 6 (Omicron variant) of the revised version of the manuscript (pages 20 and 22).

Could the word 'survival' lead to misinterpretation of the data in Figure 1? I would suggest removing 'survival' from the figure title (currently: "Figure 1 – Survival time to symptom resolution by type of treatment in the study population infected with Δ". Same for Figure 2).

We do agree that the word “survival” can be misleading. The word has been deleted from footnotes and figures.

Could Figure 1 and Figure 2 be combined into a multi-panel figure so the data are next to each other for comparison?

The previous Figure 1 and 2 have been combined into a multi-panel figure, now renamed Figure 2A/2B (without 95% CIs, as suggested by Reviewer 3).

On lines 222/223, this statement is redundant when you have presented the days of symptoms in each group "turning out to be 5 days shorter in the sotrovimab group compared to both bamlanivimab/etesevimab and casirivimab/imdevimab groups"

The statement has been removed.

I was confused about the presentation of the statistical analyses of the time to symptom resolution by treatment assignment in the two cohorts (δ and omicron). For the δ data, an analysis was presented assessing whether the duration of symptoms differed by treatment assignment (log-rank chi-square), and then each arm vs arm comparison was also presented as log-rank chi-square (lines 189-194). Cox regression was then used to confirm no difference with casirivimab/imdevimab as the reference. For omicron, where the data suggest there may be a difference, we do not present the chi-square analyses (lines 220-225). Is there a reason for this? The data should be presented the same way for both VOC.

We thank the reviewer for the comment. The Chi-square analyses on the Omicron group, which were improperly missing in the previous version of the manuscript, have been added (page 10, lines 237-242).

Lines 224-226 too strongly imply a benefit here with casirivimab/imdevimab. And could be read as suggesting both casirivimab/imdevimab and sotrovimab retain comparable activity in vitro. Of course, it depends on the assay used, etc. And these in vitro assays have significant limitations, in terms of their utility to make treatment decisions (and decisions re: guideline recommendations!). But I think this section could be re-worded to improve accuracy.

We do agree with the reviewer. The trial is underpowered to reach any firm conclusion. This major limitation has been better pointed out in the discussion of the revised version of the manuscript (page 12, lines 290-298).

When this is published, we will be on to the next variant (BA.2, BA.4, etc), in most places we already are. Therefore, consideration of how the authors frame the results in that context, ensuring we do not necessarily imply that all variants termed 'omicron' are equal with regards to mAb neutralisation, will be important (and challenging).

We thank the reviewer for the suggestion. This limitation has been pointed out in the discussion of the revised version of the manuscript (page 12, lines 298-301).

The authors could consider more of a discussion on the paucity of clinical data on these mAbs and these variants from well-designed trials, and that we do not know whether the in vitro findings have informed so many treatment/guideline decisions, do correlate with in vivo activity. Is this the first study to report clinical outcomes randomising patients across δ and omicron with these mAb agents? The authors could consider referring to this study, the other large comparative effectiveness study comparing monoclonal antibodies (particularly if published soon). https://www.medrxiv.org/content/10.1101/2021.09.03.21262551v1

We thank the reviewer for the suggested reference, which has been added in the revised version of the manuscript (page 4, lines 85-86). This study reports results as of June 25, 2021. During the trial the Α variant was the dominant variant of concern, while the Δ variant became more prevalent in the final time period. To date, our study is actually the first one to report clinical outcomes against the Omicron variant.

Reviewer #3 (Recommendations for the authors):My overall review is in the public review. What follows is details. Also, authors should check and review extensively for improvements to the use of English.Abstract– Notes AIFA and ORCHESTRA funding, but I believe the electronic form says no funding.

As stated in the manuscript, AIFA and ORCHESTRA provided funding for the study. This information has been added in the electronic form as well.

Introduction– Other comparative effectiveness trials have been conducted.https://www.medrxiv.org/content/10.1101/2021.12.23.21268244v1https://www.sciencedirect.com/science/article/pii/S1551714422001483

Thanks for the suggestion. The references have been added in the revised version of the manuscript (page 4, lines 84-87).

Methods– Please describe which centers participated and where they were located.

The information on the trial centers has been added in the revised version of the manuscript (page 5, lines 98-99).

Results– 319 pts enrolled in 5 weeks, over the winter holidays is impressive.– Leveraging of the ~50/50 δ/omicron breakdown is very nice.– For completeness, would note the #s of all the secondary outcomes, even if they are very small in # or even zero.

The numbers of all secondary outcomes have been reported in Table 4 (Δ variant) and Table 6 (Omicron variant) of the revised version of the manuscript (pages 20 and 22).

Discussion– Would note that there was a signal in only 1 secondary outcome – could note it is perhaps due to the fact that very few patients had "progression" – but that nonetheless this signal is from 1 of multiple examined outcomes

We do agree with the reviewer. The trial is underpowered to reach any firm conclusion. This major limitation has been better pointed out in the discussion of the revised version of the manuscript (page 12, lines 290-298).

– On page 10, line 255, "significant benefit" is somewhat problematic – statistically for the reason noted above, and 5d less symptoms is of debatable significance if ultimate outcome is the same. Would use more objective/neutral words.

We thank the reviewer for the suggestion. The statement has been reworded accordingly, replacing “sotrovimab showed a significant benefit” with “sotrovimab seemed to show a benefit” in the revised version of the manuscript (page 11, line 277).

– On page 11, line 265 – what are the specific findings that support that all study treatments were active against Δ (when there was no "control" arm) – the fact that no disease progression occurred – if so, pls say so?

We are sorry for the misleading sentence, which has been rephrased in the revised version of the manuscript (page 12, lines 283-290).

We fully agree with the reviewer on the limitation of the study design.

Also, what are the specific findings that support that C-I and S both retain activity against Omicron when S was superior to both C-I and B-E for symptom resolution, and C-I and B-E had the same 12d median symptom duration? Ie. what is evidence that C-I retained activity v Omicron?– Would note that currently the dominant Omicron variants are now BA.2 and beyond, and for this reason, FDA revoked all EUAs for mAbs, except for bebtelovimb.

We thank the reviewer for the suggestion. This limitation has been pointed out in the discussion of the revised version of the manuscript (page 12, lines 298-301).

Tables + Figures– Please shorten everything to 1 or 2 decimal points.

All numbers in Tables and Figures have been shortened to 2 decimal points (*p* value excluded).

– The K-M curves are hard to read – try dropping the 95% CIs.

We have provided Kaplan-Meier curves without 95% CIs (Figure 2A/2B).

[Editors' note: further revisions were suggested prior to acceptance, as described below.]

The manuscript has been improved but there are some remaining issues that need to be addressed, as outlined below:As recommended by the reviewers, please edit the abstract to ensure the conclusion reflects the study design and data. The abstract should state that the trial was ceased early and the conclusion amended as per reviewer 2 comments below.

The abstract has been revised accordingly.

For the study methods, the original sample size calculation should be included in the methods section (rather than referring readers to clinical trial register).

The original sample size calculation has been included in the method section.

Reviewer #2 (Recommendations for the authors):Thank you for submitting your revised manuscript and for addressing the suggestions raised in previous reviews. The revised manuscript is clearer, in terms of the study design and the important limitations, and the conclusions are now better supported by the data.Given the uncertainty, the limitations, etc, the conclusion of the abstract is important. I suggest a minor edit of this sentence:"Casirivimab/imdevimab seems to maintain a role in preventing COVID-19 progression in adult outpatients with early mild-to-moderate SARS-CoV-2 infection due to Omicron."Which could be changed to:"Casirivimab/imdevimab may maintain a role in preventing COVID-19 progression in adult outpatients with early mild-to-moderate SARS-CoV-2 infection due to Omicron."This is particularly true when you are not suggesting it is used for treatment (which could be implied by "seems to maintain a role") and your overarching conclusion is that adaptive clinical studies are required (although casirivimab/imdevimab is not included in the next phase of MANTICO, presumably because these data are only available now and the VOCs have continued to change).

We thank the reviewer for the suggestion. The sentence has been amended accordingly in the revised version of the abstract.

Reviewer #3 (Recommendations for the authors):Revisions are appropriate, however, manuscript revisions (esp Discussion revisions) have not extended to the Abstract.For Abstract, I recommend:1. Cut decimal places to two.

All numbers have been shortened to 2 decimal points in the revised version of the abstract (*p* value included).

2. Also add to the 2nd Conclusion sentence the specific Omicron variants analyzed in this study. Or reword so it's clear that the specific variants apply to all Conclusion sentences / mAbs.

The conclusion of the abstract has been reworded to clarify that the Omicron variants analysed in this study apply to all sentences

3. Add to Conclusion a sentence reflective of the "results are severely limited" sentence added to Discussion.E.g., could note that time to symptom resolution was 1 of X secondary outcomes, and no conclusions could be made about the primary outcome.Specific text up to the authors – my main residual concern is simply that the Abstract's Conclusion is considerably "stronger" than the manuscript's conclusion/discussion, and should be appropriately caveated.

We do agree with the reviewer. A sentence pointing out the severe limitations of the results has been added to the abstract.